# The Vicious Circle of Hepatic Glucagon Resistance in Non-Alcoholic Fatty Liver Disease

**DOI:** 10.3390/jcm9124049

**Published:** 2020-12-15

**Authors:** Katrine D. Galsgaard

**Affiliations:** 1Department of Biomedical Sciences, Faculty of Health and Medical Sciences, University of Copenhagen, 2200 Copenhagen, Denmark; katrine@sund.ku.dk; Tel.: +45-6044-6145; 2Novo Nordisk Foundation Center for Basic Metabolic Research, Faculty of Health and Sciences, University of Copenhagen, 2200 Copenhagen, Denmark

**Keywords:** autophagy, amino acids, glucagon, NAFLD, the liver–alpha cell axis

## Abstract

A key criterion for the most common chronic liver disease—non-alcoholic fatty liver disease (NAFLD)—is an intrahepatic fat content above 5% in individuals who are not using steatogenic agents or having significant alcohol intake. Subjects with NAFLD have increased plasma concentrations of glucagon, and emerging evidence indicates that subjects with NAFLD may show hepatic glucagon resistance. For many years, glucagon has been thought of as the counterregulatory hormone to insulin with a primary function of increasing blood glucose concentrations and protecting against hypoglycemia. However, in recent years, glucagon has re-emerged as an important regulator of other metabolic processes including lipid and amino acid/protein metabolism. This review discusses the evidence that in NAFLD, hepatic glucagon resistance may result in a dysregulated lipid and amino acid/protein metabolism, leading to excess accumulation of fat, hyperglucagonemia, and increased oxidative stress contributing to the worsening/progression of NAFLD.

## 1. Processing and Secretion of Glucagon

Glucagon is encoded by the preproglucagon gene (GCG) [1], expressed in alpha cells, enteroendocrine L-cells, and the brain [2]. In alpha cells, GCG is processed by prohormone convertase (PC) 2 yielding glucagon [3], whereas in L-cells, GCG is processed by PC1/3, yielding glucagon-like peptide (GLP)-1, GLP-2, glicentin, and oxyntomodulin [4]. However, recent evidence suggests that the processing is not as tissue-specific as previously thought, since PC1/3 may also be active in alpha cells, giving rise to pancreatic GLP-1 [5,6,7,8], and PC2 might act in L-cells to create gut-derived glucagon [9,10,11]. It is a matter of debate whether this only occurs as a consequence of metabolic alterations, such as pancreatectomy [9,10] and bariatric surgery [11], resulting in gut-derived glucagon, or glucagon receptor inhibition [12] and streptozotocin treatment [13], resulting in pancreatic-derived GLP-1. Pancreatic GLP-1 has been suggested to be part of heathy physiology by contributing to a local incretin axis [14,15,16]. In the measurement of glucagon, analytical challenges due to cross-reactivity with other GCG-derived peptides have been an issue [17,18], and using liquid chromatography–mass spectrometry, glucagon secretion was not found to be increased after bypass surgery [19]. Upon secretion, glucagon binds to its receptor, the glucagon receptor, and regulates hepatic glucose, lipid, and amino acid/protein metabolism [20].

## 2. The Glucagon Receptor as a Target in the Treatment of Type 2 Diabetes, Obesity, and NAFLD

In recent years, the glucagon receptor has emerged as a target in the treatment of type 2 diabetes [21,22,23,24], obesity, and non-alcoholic fatty liver disease (NAFLD) [25,26,27]. An international consensus panel recently proposed that NAFLD should be renamed to metabolic-associated fatty liver disease (MAFLD) [28]. The criteria for MAFLD are evidence of hepatic steatosis in addition to either overweight/obesity, the presence of type 2 diabetes, or evidence of metabolic dysfunction [28]. Glucagon receptor signaling increases hepatic glucose production and a majority of subjects with type 2 diabetes show increased plasma concentrations of glucagon (hyperglucagonemia) [29,30]. Hyperglucagonemia is now known to be at least partly responsible for diabetic hyperglycemia [31,32], and therefore, clinical trials using glucagon receptor antagonists (GRAs) have been conducted in subjects with type 2 diabetes. GRA treatment improves diabetic hyperglycemia [21,22,23,24], but disturbing adverse effects including increased plasma concentrations of low-density lipoprotein and liver enzymes (aspartate- and alanine-transaminase) and hepatic fat accumulation halted further development of GRAs [33,34]. However, based on the observed changes in lipid metabolism and the possibility that glucagon may decrease food intake [35,36] and increase energy expenditure [37,38,39,40] (possibly through the sympathetic nervous system [41] or other indirect effects [42]), the focus has now shifted from glucagon receptor inhibition to glucagon receptor activation.

Glucagon analogs and agonists are being investigated as potential agents in NAFLD treatment partly due to glucagon’s regulatory effects on hepatic lipid metabolism, where glucagon increases hepatic lipolysis and fatty acid oxidation [43,44] while decreasing lipogenesis [45] and the secretion of triglycerides and very-low-density lipoprotein [46,47,48]. Consistent with this, subjects with endogenous glucagon deficiency (pancreatectomized subjects) [49] and diabetic (db/db) mice treated with glucagon receptor antisense oligonucleotide [50] have increased hepatic fat. Furthermore, glucagon has a lipid-mobilizing effect and decreases plasma concentrations of cholesterol [51,52,53,54,55], non-esterified fatty acids [44], triglycerides [52,56,57], and very-low-density lipoprotein [47], suggesting that glucagon may improve dyslipidemia.

To counteract the hyperglycemic effect of glucagon, glucagon receptor agonists are being combined with agonists of either one or both of the receptors for the incretin hormones: GLP-1 and glucose-dependent insulinotropic polypeptide [58,59,60]. Glucagon/GLP-1 co-agonists reversed obesity in diet-induced obese mice, decreased body weight to a larger degree than GLP-1 agonist comparators [61], and appear to increase energy expenditure [62]. Glucagon/GLP-1 co-agonists also improve NAFLD [63] and show promising results in models of non-alcoholic steatohepatitis, a subtype of NAFLD which potentially can progress to liver fibrosis, cirrhosis, and hepatocellular carcinoma [64], by decreasing inflammation and fibrosis while increasing mitochondrial β-oxidation, thus reducing hepatic steatosis [65]. Improvement of NAFLD has also been observed in obese or overweight individuals with type 2 diabetes treated with a glucagon/GLP-1 co-agonist [66]. Glucagon’s effect on energy expenditure and satiety may be translatable to humans, since glucagon/GLP-1 and glucagon infusions, but not the GLP-1 infusion, increased energy expenditure to a similar degree in overweight subjects without diabetes [67], and a glucagon/GLP-1 infusion decreased food intake [68]. In another human study, glucagon infusion decreased food intake but had no effect on energy expenditure, possibly due the low dose used compared to other studies [69]. The effect on energy expenditure may thus be mediated by indirect effects, possibly by glucagon acting on the GLP-1 receptor, which can be activated by high concentrations of glucagon [70,71], whereas the hepatic effects of the co-agonists may be ascribed to direct glucagon receptor activation since it, unlike the GLP-1 receptor [72], is expressed in the liver.

## 3. The Liver–Alpha Cell Axis May be Impaired in NAFLD

The rationale for the use of GRAs and glucagon agonists in the treatment of patients with type 2 diabetes (55–68% of whom have liver steatosis [73]) and obesity is based on glucagon’s effects on hepatic glucose and lipid metabolism, respectively. However, glucagon is also a key regulator of amino acid/protein metabolism [74,75,76,77,78], and glucagon has been suggested to be even more important for amino acid/protein metabolism than glucose metabolism [79]. Glucagon is part of an endocrine feedback loop (the liver–alpha cell axis), in which amino acids control glucagon secretion and the proliferation of alpha cells, while glucagon in turn controls hepatic amino acid metabolism and ureagenesis [80,81,82,83,84,85].

During conditions of disrupted glucagon signaling, glucagon’s effect on amino acid metabolism is impaired and as a consequence, hyperaminoacidemia develops, which in turn results in hyperglucagonemia [80,82,83,84,86,87]. In line with this, the hyperglucagonemia observed in most subjects with type 2 diabetes is associated with hyperaminoacidemia. Especially, elevated plasma levels of alanine seem to mark a disturbed liver–alpha cell axis [85,88,89,90], and alanine also appears to be a potent glucagonotropic amino acid [91]. Importantly, hyperglucagonemia seems to be associated with a fatty liver rather than with diabetes per se [92], and plasma concentrations of glucagon and non-branched-chain amino acids are characteristically increased in subjects with increased liver fat (as reflected by elevated HOMA-IR levels) [85]. Furthermore, ureagenesis is impaired in subjects with NAFLD and non-alcoholic steatohepatitis [93,94] as well as in mice with impaired glucagon receptor signaling and obese Zucker rats with hepatic steatosis [89]. Impaired liver function thus mimics the conditions of experimental glucagon deficiency (e.g., brought about by GRAs [87]), leading to the suggestion that subjects with impaired liver function due to steatosis also show glucagon resistance [95,96]. Interestingly, subjects with NAFLD appear to exhibit resistance towards glucagon-stimulated amino acid metabolism but not towards glucagon-stimulated glucose metabolism, since the hyperglycemic effect of glucagon was preserved in a pancreatic clamp study carried out in individuals with biopsy-verified NAFLD versus lean controls, whereas the effect on amino acid turnover and ureagenesis was attenuated [97]. This is consistent with the fact that the biochemical pathway of glucagon-stimulated glycogenolysis is separate from that involved in ureagenesis, whereas gluconeogenesis is closely associated with amino acid turnover and ureagenesis [98]. Considering that glucagon mainly promotes hepatic glucose production via glycogenolysis [99], this may explain why sensitivity to glucagon’s hyperglycemic effect can be preserved while its hypoaminoacidemic effect is impaired. In contrast, rats with high-fat-diet–induced hepatic steatosis showed a reduction in glucagon-stimulated hepatic glucose production, most likely due to reduced glycogenolysis [100], and this state of glucagon resistance could be reversed by exercise training, which resulted in a reduction in hepatic liver triglycerides [100]. Studies by the same group also showed that hepatic triglyceride accumulation (induced by a high-fat diet) reduced hepatic glucagon receptor density and glucagon-mediated signal transduction [101,102]. Whether this applies to human liver steatosis is unknown.

## 4. Hepatic Glucagon Resistance May Impair Autophagy Resulting in Increased Oxidative Stress

Exogenous glucagon stimulates hepatic autophagy [103,104,105,106], and during conditions of increased endogenous glucagon secretion, such as starvation [107], hypoglycemia [108], and type 1 diabetes [109], the number of hepatic lysosomes is increased. By stimulating autophagic protein degradation, glucagon may provide substrates for gluconeogenesis and ketogenesis and thus, sustain nutrient and energy homeostasis during starvation [110,111]. Autophagy also functions to remove damaged and non-functional organelles [112], and the autophagic degradation of mitochondria (mitophagy), which may be induced by glucagon [113], is important for maintaining mitochondrial turnover. Impaired mitophagy thus leads to the accumulation of dysfunctional mitochondria generating reactive oxygen species, resulting in oxidative stress and inflammation [114]. Autophagy is, furthermore, an important regulator of lipid metabolism and lipophagy; in other words, the degradation of triglycerides into free fatty acids in autolysosomes functions to regulate intracellular lipid stores and energy homeostasis by utilization of the released free fatty acids in mitochondrial β-oxidation [115]. Inhibition of autophagy in cultured hepatocytes and mouse livers increased triglyceride storage in lipid droplets and decreased β-oxidation and very-low-density lipoprotein secretion [116], processes which are influenced in the opposite direction by glucagon [43,44,56]. The hepatic lipolytic effect of glucagon could therefore potentially be partly mediated by lipophagy. The process of autophagy has been shown to be impaired in obesity and NAFLD [117], and hepatic resistance to glucagon in subjects with NAFLD may therefore partly explain the impaired autophagy. Amino acids, especially glutamine, leucine, and arginine, are potent activators of the mammalian target of the rapamycin complex 1 [118], which suppress autophagy [119]. The hyperaminoacidemia observed in NAFLD may therefore further impair hepatic autophagy. In NAFLD, impaired hepatic autophagy would result in the worsening/progression of NAFLD by increasing oxidative stress and lipid accumulation, and increased glucagon signaling could potentially improve NAFLD by increasing the autophagic removal of damaged organelles and lipid droplets. Whether the stimulatory effect of glucagon on hepatic autophagy is a contributing factor to the reversal of NAFLD observed by glucagon agonist treatment is currently unknown; however, the possibility provides a new perspective on the role of tri- and dual-agonists in the treatment of NAFLD.

In summary, hepatic glucagon resistance may result in a dysregulated lipid and amino acid/protein metabolism and possibly impaired autophagy, leading to excess accumulation of fat, increased oxidative stress, and hyperaminoacidemia, resulting in hyperglucagonemia and hyperglycemia. Hepatic glucagon resistance in NAFLD would thus result in the creation of a vicious circle and contribute to the worsening/progression of NAFLD (Figure 1 and Table 1). However, the majority of studies investigating these aspects of NAFLD have been performed on cells and rodents, some with contrasting findings. Further studies translating these observations to humans are thus warranted. Recognizing that glucagon’s primary role may not reside in glucose metabolism alone, but also includes important aspects of lipid and amino acid/protein metabolism, is becoming increasingly important in view of the current development of glucagon-based therapeutics for the treatment of NAFLD and obesity.

## Figures and Tables

**Figure 1 jcm-09-04049-f001:**
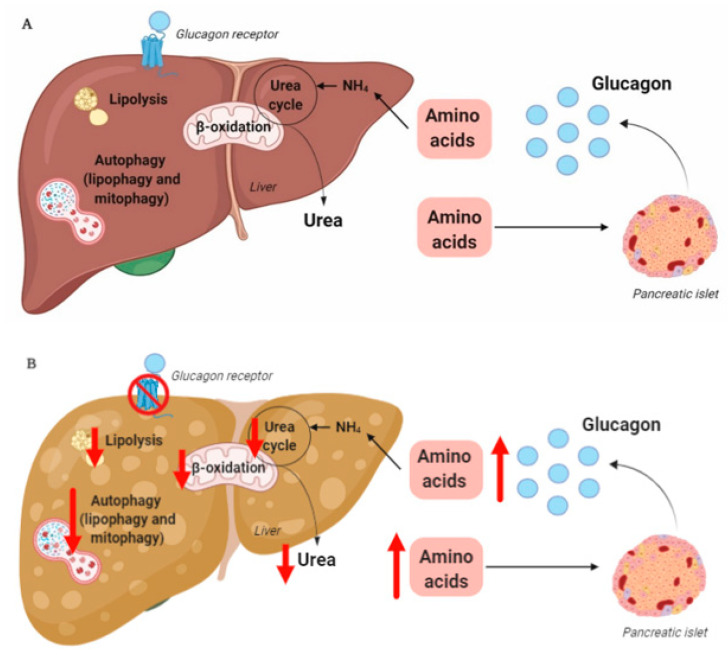
(**A**) Glucagon is secreted from the pancreatic alpha-cell and binds to its hepatic receptor. This results in increased hepatic amino acid uptake and metabolism (ureagenesis). Hepatic glucagon signaling also increases lipolysis and ß-oxidation and may increase hepatic autophagy. (**B**) During conditions of disrupted glucagon signaling, hepatic amino acid metabolism and ureagenesis is halted, as is hepatic lipolysis, ß-oxidation, and autophagy. This results in the accumulation of hepatic fat and damaged organelles, inducing oxidative stress and increased plasma amino acids, leading to hyperglucagonemia. This figure was created using BioRender.

**Table 1 jcm-09-04049-t001:** The molecular mechanisms that may be impaired by hepatic glucagon resistance inhibition and the pathologies that are involved.

	Metabolic Process	Pathology
Amino acid/protein metabolism	Amino acid transport Amino acid catabolism Ureagenesis	Hyperaminoacidemia Hyperglucagonemia Hyperammonemia
Autophagy	Lipophagy Mitophagy	Increased hepatic fat Increased oxidative stress
Lipid metabolism	β-oxidation Lipolysis	Increased hepatic fat Dyslipidemia

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
