# Peer review of "The Vicious Circle of Hepatic Glucagon Resistance in Non-Alcoholic Fatty Liver Disease"

_jcm, 2020, doi:10.3390/jcm9124049_

Round 1

Reviewer 1 Report

The review “The vicious circle of hepatic glucagon resistance in non-alcoholic fatty liver disease” summarizes the information regarding glucagon resistance and its effects on metabolism trough an ample bibliography revision. In addition, other manuscripts signed by this author explain other aspects of the relevance of glucagon.

The reviewer finds the manuscript very interesting and the topics that it touches very adequate. However, I would like to make some minor comments for the improvement of this publication.

General aspects of the Manuscript:

I would find the manuscript more readable if some titles are added to each paragraph, and/or if paragraphs are smaller... This could help the reader to follow better all the information given.

The bibliography references are adequate for this publication, including poster communications, and old and new papers. Though, some of them talk about the same topics, 115 appear to be a high number considering the length of the review.

Punctual aspects of the Manuscript:

In the 3rd line of the abstract says "ranging from simple steatosis to cirrhosis"... Most of experts in NAFLD agree that this disease ranges from simple steatosis to NASH, and that NASH may progress to cirrhosis and hepatocellular carcinoma. "The subtype of NAFLD that is histologically categorised as non-alcoholic steatohepatitis (NASH) has a potentially progres- sive course leading to liver fibrosis, cirrhosis, hep- atocellular carcinoma (HCC) and liver transplantation". Ref: Journal of Hepatology 2019 vol. 70 j 531–544. It is also relevant to note that NAFLD has different pathophysiological manifestations depending on the BMI of the patient.

The referee misses more concrete information regarding lipid and amino acid species. In fact some lipids are key responsible for VLDL assembly and secretion in the liver, such as phosphatidylcholines, and, as an idea they might be afected by glucagon-induced endoplasmic reticulum stress. Ref:Biochem. Soc. Trans. (2014) 42, 1447–1452; doi:10.1042/BST20140138. In addition, some amino acids per se stimulate or inhibit autophagy. Ref:Autophagy. 2018;14(2):207-215. doi: 10.1080/15548627.2017.1378838. Epub 2017 Dec 31.

It is logical that the majority of novel studies have been performed on cells and rodents, and researchers must know the difficulties in translating these aseverations to humans. Throughout the text this comments are made, but they should be stated in the final summary as well, together with comments on the divergences found in different studies.

Figure 1 is very helpful to understand the mechanisms that are commented in the manuscript. I would add that the secretion of VLDL from the liver may be decreased after glucagon receptor inhibition.

I would also strongly recomment to add a Table with a summary of the molecular mechanisms that are affected by alterations in the concentration of glucagon and/or inhibition of glucagon receptor and the pathologies that are involved.

I hope that my humble comments are useful for the authors and for the editors.

Best wishes,

The reviewer

Reviewer 2 Report

This is a concise and well-written review, discussing the importance of glucagon in the pathogenesis of NAFLD (with the current consensus to be called as “MAFLD”, metabolic dysfunction-associated fatty liver disease (Journal of Hepatology 2020 vol. 73 j 202–209).

I have only few suggestions to the author:

  1. From the figure it is not clear what happens to Urea in the condition B. An arrow, and more explicit description of the Figure legend would help the better understanding of the reader.
  2. The author may want to consider to change the NAFLD into MAFLD. At least a mention of the new consensus should be made.

Author Response

  • I agree that what happens to urea in condition B of figure 1 is not clear. In the revised manuscript, I have added an arrow to the figure and modified the figure legend as shown below.

Figure 1. (a) Glucagon is secreted from the pancreatic alpha-cell and binds to its hepatic receptor. This results in increased hepatic amino acid uptake and metabolism (ureagenesis). Hepatic glucagon signalling also increases lipolysis and ß-oxidation and may increase hepatic autophagy. (b) During conditions of disrupted glucagon signalling hepatic amino acid metabolism and ureagenesis is halted as is hepatic lipolysis, ß-oxidation, and autophagy. This results in accumulation of hepatic fat and damaged organelles inducing oxidative stress and increased plasma amino acids leading to hyperglucagonemia. This figure was created using biorender.

  • I agree that the new consensus metabolic dysfunction-associated fatty liver disease (MAFLD) should be mention, thank you for bringing this to my attention. I have added a paragraph on MAFLD as shown below.

Page 2 line 41-44: An international consensus panel recently proposed that NAFLD should be renamed to metabolic-associated fatty liver disease (MAFLD) [37]. The criteria for MAFLD is evidence of hepatic steatosis in addition to either overweight/obesity, presence of type 2 diabetes, or evidence of metabolic dysfunction [37].

Reviewer 3 Report

Katrine Galsgaard reviews the emerging and significant relationship between glucagon and liver steatosis in this submitted article.

It is a timely exploration of this subject and is executed to a high standard.

A number of suggestions to increase its excellence are are follows:

Could the review be re-structured to discuss Glucagon, Glucagon receptor agonists and antagonists in Diabetes and then discuss the combination of GLP-1 and GIP with glucagon? Currently, much of the discussion of combination agent precedes the discussion on Glucagon receptor activation/antagonism example lines 36-52.

Could discussion of the preproglucagon gene and processing differences between alpha cells and L-cells discussed? Filip Knop's work discussing the extra-islet courses of glucagon should also be discussed in more detail.

Where ever possible additional description of the signaling pathways linking the glucagon receptor vs. indirect signaling should be discussed and differentiated. 

Author Response

Dear reviewer, thank you for your constructive feedback.

  • I agree that the review is not structured as well as it could be in regards to glucagon, glucagon receptor agonists and antagonists in Diabetes. I have now restructured the manuscript so that this is discussed in line 56-67, preceding the discussion on combination agents in line 68-84.
  • In the revised manuscript, I have inserted a paragraph discussing the preproglucagon gene and processing differences between alpha cells and L-cells including the extra-islet courses of glucagon. The paragraph is shown below.

Page 1 line 25-38: Glucagon is encoded by the preproglucagon gene (GCG) [1], expressed in alpha cells, enteroendocrine L-cells [2], and neurons of the nucleus tractus solitarius [3]. In alpha cells GCG is processed by prohormone convertase (PC) 2 yielding glucagon [4], whereas in L-cells GCG is processed by PC1/3, yielding glucagon-like peptide (GLP)-1, GLP-2, glicentin, and oxyntomodulin [5]. However, recent evidence suggest that the processing is not as tissue specific at previously thought, since PC1/3 may also be active in alpha cells giving rise to pancreatic GLP-1 [6-9] and PC2 might act in L-cells to create gut derived glucagon [10-13]. It is a matter of debate whether this only occurs as a consequence of metabolic alterations, such as pancreatectomy [10,11] and bariatric surgery [13], resulting in gut derived glucagon, or glucagon receptor inhibition [14,15] and streptozotocin treatment [8,16], resulting in pancreatic derived GLP-1. Pancreatic GLP-1 has been suggested to be part of heathy physiology by contributing to a local incretin axis [17-19]. In the measurement of glucagon, analytical challenges due to assay cross reactivity with other GCG derived peptides has been an issue [20,21], and using liquid chromatography-mass spectrometry the gut was not found to secrete glucagon after bypass surgery [22].

  • This is a good point. In the revised manuscript, I have differentiated between direct and indirect glucagon receptor signaling as shown below.

Page 3 line 81-84: The effect on energy expenditure may thus be mediated by indirect effects, possibly by glucagon acting on the GLP-1 receptor which can be activated by high concentrations of glucagon [90,91], whereas the hepatic effects of the co-agonists may be ascribed to direct glucagon receptor activation since it, unlike the GLP-1 receptor [92], is expressed in the liver.

Reviewer 4 Report

The manuscript deals with an important issue of glucagon resistance, and its emerging role in NAFLD, which nowadays is increasingly seen as the most important chronic liver disease.
The author has provided an extensive overview of available data on physiological roles of glucagon and glucagon receptor, revealing glucagon as a key regulator of amino acid/protein metabolism. The roles of Glucagon Receptor Antagonists and Glucagon agonists in the treatment of patients with type 2 diabetes, obesity, and liver steatosis are also discussed.
Summarizing the abundance of cited references the author suggests that “glucagon´s primary role may not reside in glucose metabolism alone, but also includes important aspects of lipid and amino acid/protein metabolism”. It is of the utmost importance to view the current developments of glucagon based therapeutics as a possibility to recognize its potential role for NAFLD and obesity treatment.

Author Response

Thank you very much for your positive comments.

This manuscript is a resubmission of an earlier submission. The following is a list of the peer review reports and author responses from that submission.